# Systematic Propagation of AVHRR AOD Uncertainties—A Case Study to Demonstrate the FIDUCEO Approach

**Thomas Popp [1],*** **and Jonathan Mittaz [2,3]**

[1] Deutsches Zentrum für Luft- und Raumfahrt e. V. (DLR), Deutsche Fernerkundungsdatenzentrum (DFD), 82234 Wessling, Germany
[2] Department of Meteorology, University of Reading, Reading RG6 6AH, UK; j.mittaz@reading.ac.uk
[3] National Physical Laboratory, Teddington TW11 0LW, UK
***** Correspondence: thomas.popp@dlr.de; Tel.: +49-8153-28-1382

**Abstract:** The AVHRR aerosol optical depth (AOD) is inverted from measured reflectances in the red band using a statistical correlation of surface reflectance with mid-infrared channel reflectances and a modelling climatology of the aerosol type. For such a sensor not specifically designed for AOD retrieval, propagating uncertainties is crucial because the sensitivity of the retrieved AOD to the measured signal varies largely with retrieval conditions (AOD itself, surface brightness, aerosol optical properties/aerosol type, observing geometry). In order to quantify the different contributions to the AOD uncertainties, we have undertaken a thorough analysis of the retrieval operator and its sensitivities to the used input and auxiliary variables. Uncertainties are then propagated from measured reflectances to geophysical retrieved AOD datasets at the super-pixel level and further to gridded daily and monthly products. The propagation uses uncertainty correlations of separate uncertainty contributions from the FIDUCEO easyFCDR level1b products (common fully correlated, independent random, and structured parts) and estimated uncertainty correlation structures of other major effects in the retrieval (surface brightness, aerosol type ensemble, cloud mask). The pixel-level uncertainties are statistically validated against true error estimates versus AERONET ground-based AOD measurements. It is shown that a 10-year time record over Europe compares well to a merged multi-satellite record and that pixel-level uncertainties provide a meaningful representation of error distributions. The study demonstrates the benefits of new recipes for uncertainty characterization from the Horizon-2020 project FIDUCEO ("Fidelity and uncertainty in climate data records from Earth Observations") and extends them further with recent additions developed within the ESA Climate Change Initiative.

**Keywords:** uncertainty propagation; aerosol optical depth; satellite-based climate data records



## 1. Introduction

Within the ESA's Climate Change Initiative (CCI) [1], the provision of uncertainties for each measurement was established as a standard for its satellite-based climate data records; in-depth discussions for the large set of different essential climate variables led to common standards for uncertainties in the products as far as suitable [2]. It was also established that a thorough quantification of uncertainties in the climate data records is essential for assessing the consistency of different variables and long-term records based on a series of similar instruments [3]. The discussion also revealed that the common standards for uncertainties in (laboratory) measurements cannot be directly applied to all satellite-based datasets, since the inversion steps to derive the thematic climate data records (geophysical variables) include the use of auxiliary and climatological datasets.

This finding led to close collaboration of Earth observation specialists with metrological experts to adapt the common standards of the GUM (guide for uncertainties in measurements [4]) for the satellite retrieval products [5]. An adapted fundamental concept for rigorous propagation of uncertainties in satellite-based climate data records was

developed, "the FIDUCEO approach". This starts from a systematic analysis of the measurement equation to identify all relevant contributions to the product uncertainties and then implements a standard approach to document the key characteristics of each effect in the measurement equation (including the maturity of their quantitative understanding and their spatio-temporal correlations).

Aerosols in the atmosphere are of global importance for the climate system by reflecting solar radiation back into space (and in case of dark particles, also regionally absorbing solar radiation) and by acting as cloud condensation nuclei, which again reflect solar radiation from the Earth. Aerosol observation from space is typically an ill-posed problem that entails significant uncertainties for the derived data records. Concretizing the standards of [2], the CCI standards for the derivation and characterization of aerosol climate data records were developed [6,7] which also included the requirement that pixel-level uncertainties themselves need to be validated. Methodologies for evaluating the propagated uncertainties were then demonstrated in [8].

This study demonstrates the application of the FIDUCEO recipe (following [5]) to establish rigorous uncertainty propagation for an example processing chain of a thematic climate data record for aerosols based on an advanced fundamental climate data record from the AVHRR sensor. This underlying easyFCDR dataset was also developed by applying the FIDUCEO recipe to the basic sensor reflectance and brightness temperature measurement processing [9]. We apply the FIDUCEO recipe for the analysis of the major uncertainty sources in the processing chain, the development of the uncertainty estimation at pixel level and the propagation of uncertainty contributions with different correlation structures to higher aggregated product levels. We use the nomenclature from [5]. For the validation of uncertainties, we follow the best practices of [8], which we developed further by adding a quantitative analysis of error histograms derived from the uncertainty distributions.

## 2. Applying the FIDUCEO Analysis Method to AVHRR AOD Uncertainties

### 2.1. Overview of the Simple AVHRR AOD Retrieval Method

The advanced very-high-resolution radiometer (AVHRR) is a rather weak instrument for retrieving aerosol climate data records. On the one hand, it lacks on-board calibration capabilities in the visible and has only few and spectrally broad channels, which limit the information content over land for aerosols to just the total loading. On the other hand, the satellite overpass times shift significantly during and between the lifetimes of the different platforms. However, the series of AVHRR instruments offers the potential for a long-term time series dating back to the early 1980s. To make those time series valuable, a proper quantitative characterization of their uncertainties is mandatory.

The measured signal $R_{\text{TOA}} = \frac{\pi}{\mu_0} \frac{L}{E_0}$ (with $\mu_0 = \cos(\theta_0)$, and $\theta_0$ solar zenith angle, $L$ calibrated radiance, $E_0$ solar irradiance) is proportional to the scattering by aerosols (and molecules) in the atmospheric column of height $z_{\text{maxmaxm}} = 100$ km. Atmospheric aerosol loading is typically described by the aerosol optical depth, AOD, defined as the vertical integral of aerosol extinction $\sigma_e$: $\text{AOD} = \int_0^{z_{\text{max}}} \sigma_e(z) \, dz$. However, this signal contribution is also proportional to two more properties of the atmospheric aerosol, namely the single scattering albedo $\omega_0$ (which gives the scattering part of the aerosol extinction and is defined as $\omega_0 = \frac{\sigma_e - \sigma_a}{\sigma_e}$, with $\sigma_e$ extinction coefficient and $\sigma_a$ absorption coefficient) and the phase function $p(\psi)$ of the scattering angle $\psi$ for the observing geometry ($\psi = \cos^{-1}(\sin\theta_0 \sin\theta_S \cos\Delta\varphi - \cos\theta_0 \cos\theta_S)$), which quantifies the directional dependence of scattering. Furthermore, the measured signal is "disturbed" (often dominated) by a second term, which is proportional to the surface albedo $Alb_{\text{surf}}$ and the exponential weakening by extinction through aerosols (but also molecules and gases). Together, this can be described as

$$R_{TOA} \sim \text{AOD} \, \omega_0 \, p(\Psi) + Alb_{\text{surf}} \exp\left(-\frac{AOD}{\mu_0}\right) \tag{1}$$

Molecule (Rayleigh) scattering can be described analytically as a function of surface elevation/pressure. Due to their relatively low concentrations, trace gas contributions to

scattering are negligible, but they may exhibit significant absorption; however, due to the choice of radiometer window channels in spectral parts outside the major gas absorption lines, their impact on the signal is small. Both the atmospheric scatter term and the surface term interplay through multiple scattering. The inversion of AOD from measurements of $R_{TOA}$ then provides the satellite AOD retrieval, which is typically mathematically ill-posed, i.e., the number of linear independent observables is less than the number of parameters needed to describe the abundance and characteristics of atmospheric aerosol and the underlying surface. Therefore, the retrieval needs to work with intelligent assumptions/auxiliary information, which bring additional sources of uncertainties adding to the propagated uncertainties of the calibrated reflectances $R_{TOA}$. AOD is retrieved in the red band (centered at 630 nm) and converted to the commonly used mid-visible reporting wavelength of 550 nm (once the aerosol size distribution is specified as part of the aerosol type a unique conversion factor between AOD at different wavelengths is defined).

Through this interplay of the atmospheric and the surface term, the measured signal $R_{TOA}$ is a non-linear function of both $AOD_{550}$ and $Alb_{surf}$; the function depends on the observing geometry between satellite and solar position and the optical properties of the atmospheric aerosol (named aerosol type). For the retrieval of AOD over land from AVHRR single-channel measurements, the poor radiometric calibration poses a first difficulty, which can lead to slightly negative AOD values in the lowest processing level L2A (which are allowed to avoid breaking the distribution of AOD values artificially at the wrongly calibrated zero value). The second major challenge lies in the estimation of directional surface albedo, so that in the inversion the signal contribution from aerosols and from the surface can be separated. This is still the largest challenge for this simple method (and needs more work beyond this demonstration case study). The third challenge lies in propagating uncertainties through the processing levels while being able to take into account their different correlation structures—here, the FIDUCEO easyFCDR L1B dataset [9] used as input provides all necessary information to perform this propagation rigorously.

The dark field inversion approach uses a simplified approach from AATSR [10], where the main functional dependence of AOD for a selected observing geometry and for a selected aerosol type is parameterized by stepwise second-order polynomials in $R_{TOA}$ interpolated linearly between two discrete values of $Alb_{surf}$. The coefficients are pre-calculated in look-up tables with accurate forward radiative transfer calculations for a distinct grid of values of $Alb_{surf}$ (0.05 . . . 0.95 in steps of 0.005), for a distinct grid of the three observing angles $\theta_S$, $\theta_0$, $\Delta\phi$ (5° zenith angles, 10° azimuth angle) and for a distinct set of 36 pre-defined aerosol types, which are meant to describe the range and variability of the optical characteristics of aerosol in the atmosphere. To minimize the surface contribution, a so-called "dark field method" is applied, where only pixels with low $Alb_{surf}$ are exploited, since the signal sensitivity to the aerosol content decreases with increasing $Alb_{surf}$.

A surface albedo estimation method is applied (adapted from [11]), which relies over land on a correlation of surface reflectances between the channels at 3.7 μm and 630 nm. In order to estimate the surface reflectance at 630 nm, the correlation between the reflectances at 3.7 μm and 630 nm was analyzed and validated by a database of atmospherically corrected AATSR datasets (using adjacent AERONET stations to prescribe AOD). Evidently, this surface-brightness-estimation approach cannot be used over bright surface (e.g., deserts), but it is applicable over fully and partly vegetation covered surfaces. Therefore, a threshold of $Alb_{surf}^{630} < 0.075$ is used to avoid dark fields with low sensitivity to aerosol optical depth (partly vegetated dark fields are searched with $R_{TOA}^{3.7} < 0.14$ and NDVI (Normalized Differential Vegetation Index) > 0.2). Bi-directionality treatment over land is performed by use of a fixed indicatrix for the entire year (normalized BRDF which is then adjusted to each pixel brightness) "FOREST" based on airborne measurements. Note that the reflectance at 3.7 μm needs to be calculated from the brightness temperatures at 3.7 μm (AVHRR channel 3B) using the temperature of the 11 μm band (AVHRR channel 4) to determine the thermal radiance.

A linear parameterization was deduced from the AATSR atmospherically corrected datasets, which allows estimating the surface albedo at 630 nm from top of atmosphere reflectance at 3.7 μm and the NDVI. Comparing this parameterization to the exact surface albedo values yields an albedo difference histogram with a width of ~0.01, which is used as the uncertainty of the estimated surface albedo. The quantities that are used to estimate $Alb_{surf}^{630}$ are assumed independent of AOD. This is true only in the first order. At 3.7 μm, most aerosol types have only very small extinction; for large particles and large AOD, a remaining effect can no longer be neglected. The NDVI as a ratio of two channels also provides a first-order correction of its AOD dependence, but the aerosol signal at the two wavelengths 630 nm and 870 nm is different for many aerosol types. Therefore, an iterative correction for AOD impact on $R_{TOA}^{3.7}$ and NDVI is conducted.

Within Aerosol_cci, an analysis of different aerosol component definitions by the various algorithm teams and also by external (e.g., NASA teams) was made, and a simple common definition of four basic components was agreed upon [6,12] that reflects the expected information content of mid-visible retrievals. In terms of absorption, the two fine-mode components are extremes where reality can be described as a combination of these two components. For coarse mode aerosols, two alternatives, sea salt and mineral dust, are considered. Analyzing AERONET ground-based sun photometer data, the most frequent effective radius was determined near 0.14 μm for the fine mode and near 1.94 μm for the coarse mode. A climatology of monthly averaged mixing fractions between those four components is derived from an ensemble of models (AEROCOM community) and AERONET ground-based sun photometer measurements (to determine aerosol absorption) [13]. From this climatology, the mixing fractions of total $AOD_{550}$ between fine and coarse mode, within the fine mode (less absorbing of total fine mode) and within the coarse mode (dust of total coarse mode) are then used for monthly 1-degree grid cells to determine the most likely aerosol type.

Based on the four basic Aerosol_cci components, 36 mixtures with typically bi- or tri-modal size distributions are defined to cover the range and variability of realistic aerosol compositions in the atmosphere [11]. Calculating optical aerosol component properties relies on Mie calculations (i.e., spherical particles are assumed) for the fine mode and sea salt, whereas mineral dust optical properties were calculated with a T matrix (thus taking non-spherical patterns with an assumed shape) into account. The stratospheric aerosol is fixed at 0.01 $AOD_{550}$ with a stratospheric background component; the free troposphere AOD above the boundary layers is fixed at 0.02 with a tropospheric background component (sulfate only). The treatment of relative humidity in the fine mode is implicit through using two components with different absorption which cover the range of realistic values by mixing water into the particles; however, the size is not altered.

Cloud masking is a key pre-requisite of any aerosol retrieval algorithm since any misclassifications of clouds as cloud-free (and vice versa) may lead to significant errors of retrieved AOD. For this purpose, a combination of threshold tests applied to all bands from the visible over the mid-infrared to the thermal infrared spectral range combined with spatial coherence tests and using Bayesian statistics has proven most suitable (APOLLO_NG, [14]). Its result is a pixel level cloud probability value. Using a low-probability threshold (15%) guarantees a minimum cloud contamination.

A demonstration was implemented, where a regional AVHRR AOD Climate Data Record (CDR) of 10 years (2003–2012) over land covering Europe and Northern Africa was produced [15]. The processing chain consists of different levels. The major inversion of the input FIDUCEO easyFCDR Level1B $R_{TOA}$ product into Level2A AOD results is performed on single pixels, but only for an automatically selected best-suited subset of them (cloud-free dark fields). From these, aggregated super-pixels (3 × 3 pixel cells) in sensor projection are averaged to provide the basic AOD product (Level 2B). Further aggregation to gridded cells (1-degree latitude, longitude, daily and monthly) is finally achieved with level 3 processing.

*2.2. Analysis of the Simple AVHRR AOD Retrieval*

The **measurement Equation** (2) shows how AOD can be inverted from top-of-atmosphere (TOA) reflectance measurements in the red band, (directional) surface albedo estimated from the mid-infrared channel at 3.7 μm and by assuming optical properties of atmospheric aerosol (aerosol type)—note the colour coding used throughout this manuscript to identify the dominant effects:

$$\text{AOD}_{630} = g\left(R^{630}_{\text{TOA}}; \theta_S, \ \theta_0, \Delta\varphi; \ Alb_{\text{surf}}, \ aerosol_{\text{type}}\right) + 0 \tag{2}$$

where $\text{AOD}_{630}$ is the resulting aerosol optical depth at 630 nm;

$g$ is the retrieval operator;
$R^{630}_{\text{TOA}}$ is the input top-of-atmosphere reflectance at 630 nm (channel 1);
$\theta_S, \ \theta_0, \Delta\varphi$ are the observation angles (sun and observer zenith, relative azimuth);
$Alb_{\text{surf}}$ is the (directional) surface albedo;
$aerosol_{\text{type}}$ is a combination of aerosol optical properties;
and 0 is all other dependencies which on average are minor.

$R^{630}_{\text{TOA}}$ is the measured AVHRR channel 1 reflectance; $Alb_{\text{surf}}$ is estimated from the reflectance part of the AVHRR channel 3B (using a linear conversion determined by the vegetation index). The $aerosol_{\text{type}}$ is provided by a climatology (1-degree latitude/longitude, monthly) of mixing factors of four basic aerosol components (fine-mode weakly absorbing, fine-mode strongly absorbing, desert dust, sea salt). Look-up tables (LUT) of radiative transfer calculations are stored as second-order polynomials for each of 36 aerosol mixtures representing a realistic range of true atmospheric aerosol compositions. All other contributions which, on average, are considered as minor are notified in the 0-term.

Figure 1 shows the **FIDUCEO traceability chain** of this processing. AOD is retrieved in the red band (630 nm, AVHRR channel 1) for selected single "dark pixels" of the AVHRR GAC product of $4 \times 4 \ \text{km}^2$ (internal L2A product) and then aggregated to super pixels (3 $\times$ 3) of about $12 \times 12 \ \text{km}^2$ (L2B product); additional gridded products (L3) on 1-degree latitude/longitude are produced by averaging all super-pixels per day and then all daily values during a month. The main input is from AVHRR channel 1, but other channels are also needed for calculating $NDVI = \frac{R^{870}_{\text{TOA}} - R^{670}_{\text{TOA}}}{R^{870}_{\text{TOA}} + R^{670}_{\text{TOA}}}$, estimating the surface albedo, and for cloud mask tests. Dark pixels are determined (upper-left branch in the block diagram) by cloud masking (to avoid cloud contamination) and by filtering all cloud-free pixels for (partial) vegetation cover and for darkness in the mid-infrared band. AOD is retrieved by inversion according to the measurement Equation (2) (central right branch in the block diagram). In order to model the dependence of the results to different aerosol types, the retrieval is repeated for an ensemble of 36 realizations, and the most likely aerosol type is extracted from a climatological modelling dataset. For each process, the input parameters (from the L1B product variables or from interim output of a predecessor step) are shown—this introduces possible sources of uncertainty to be propagated through each step and introduces auxiliary input (aerosol component database, aerosol type climatology).

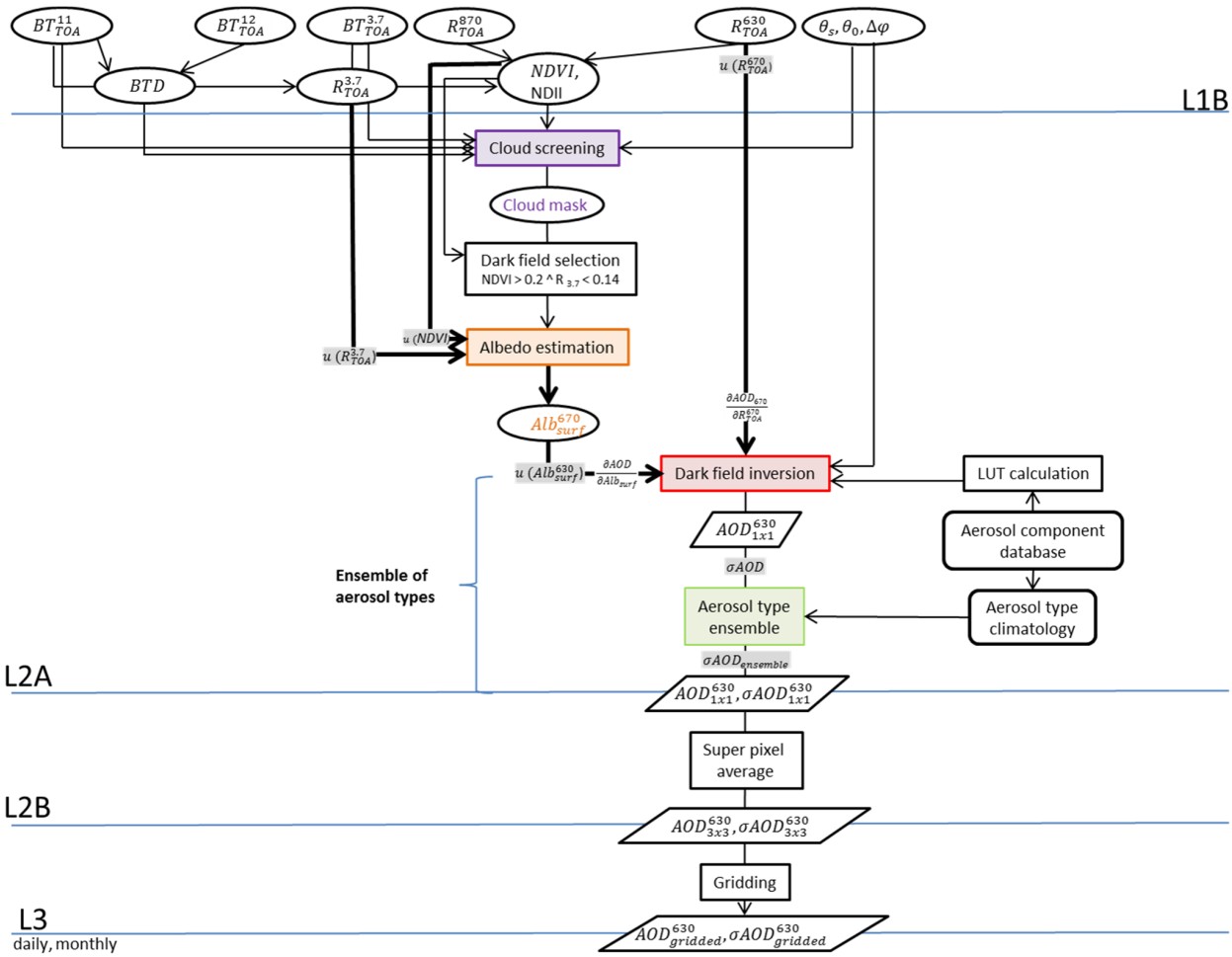

**Figure 1.** Traceability chain for AVHRR AOD CDR.

*2.3. Estimating Single Measurement Uncertainties*

Estimating uncertainties of the retrieved AOD on pixel-level $u(\text{AOD})$ is crucial to understand the reliability of the results. This is particularly important since the sensitivity of the retrieved AOD to the measured signal varies largely with retrieval conditions (AOD itself, surface brightness, aerosol optical properties/aerosol type, observing geometry). Given the weak retrieval from AVHRR, we choose a pragmatic approach for the estimation of pixel-level AOD uncertainties, which is based on lessons learned during the ESA Aerosol_cci project and focuses on the **uncertainty equation with dominant terms** (also called effects):

$$u(\text{AOD}) = \sqrt{\left(\frac{\partial \text{AOD}}{\partial R_{\text{TOA}}}\, u(R_{\text{TOA}})\right)^2 + \left(\frac{\partial \text{AOD}}{\partial Alb_{\text{surf}}}\, u(Alb_{\text{surf}})\right)^2 + \left(u(\text{AOD})_{ensemble}\right)^2 + u^2(0)} \tag{3}$$

where $u(\text{AOD})$ is the AOD uncertainty,

$\frac{\partial AOD}{\partial R_{\text{TOA}}}$ is the sensitivity of AOD to $R_{\text{TOA}}$,

$u(R_{\text{TOA}})$ is the uncertainty of $R_{\text{TOA}}$,

$\frac{\partial AOD}{\partial Alb_{\text{surf}}}$ is the uncertainty of $Alb_{\text{surf}}$,

$u(Alb_{\text{surf}})$ is the uncertainty of $Alb_{\text{surf}}$,

$u(\text{AOD})_{ensemble}$ is the spread of an ensemble of different aerosol types,

and $u^2(0)$ is the sum of weaker or ill-defined uncertainties, considered significantly smaller or assumed to be fully independent, so that they average out on larger spatial/temporal scales.

Neglected uncertainties summarized in the term $u^2(0)$ do contain trace gas absorption correction (small due to setup of window channels), altitude-dependent Rayleigh scattering correction, vertical layering of AOD (both small in the red band), look-up table errors versus full radiative transfer calculations, including interpolation errors between distinct angular values (both proven small with full radiative transfer calculations), and interpolation values between distinct aerosol types.

The uncertainty Equation (3) shows that the uncertainties in the AOD CDR not only originate from propagation of uncertainties in measured reflectances but assumptions, simplifications, and lacking knowledge in the retrieval also add major contributions. For applying Equation (3), we assume that the effects are independent of each other; we note that there may be some correlation between $R_{\text{TOA}}$ and $\text{Alb}_{\text{surf}}$ due to the use of the NDVI which has a common channel ($R_{630}$), but further complex studies are needed to quantify the consequences.

To derive AOD uncertainties, the following **FIDUCEO uncertainty tree** for the single pixel AOD inversion (L2A) was prepared (Figure 2), which determines how each of the uncertainty components can be calculated. In this diagram, the calculation of the uncertainty of each of the three dominant terms of the right side of the measurement function in the center is depicted. The (red) uncertainty of the reflectance inversion is derived as product of the reflectance uncertainty $u(R_{\text{TOA}})$ and the AOD sensitivity to the measured reflectance $\frac{\partial \text{AOD}}{\partial R_{\text{TOA}}}$. Additionally, the (brown) uncertainty of the surface albedo is calculated as a product of the albedo uncertainty $u(\text{Alb}_{\text{surf}})$ and the AOD sensitivity to the albedo $\frac{\partial \text{AOD}}{\partial \text{Alb}_{\text{surf}}}$. However, in this case, the albedo uncertainty needs to be calculated by propagating the uncertainties of NDVI (and ultimately $R_{630}$ and $R_{870}$) and $R_{3.7}$ through the linear conversion function used. Furthermore, a constant global uncertainty value of 0.01 is added which reflects the uncertainty of using the linear regression. The aerosol type uncertainty (green) cannot be calculated with a similar product, but this is replaced by the spread of an ensemble of 36 different aerosol mixtures.

Those uncertainties are calculated based on the reflectance uncertainties contained in the **FIDUCEO easyFCDR** AVHRR L1B product. This easyFCDR product provides **three separate uncertainty components** for each channel reflectance (or brightness temperature):

- **common** (globally fully correlated uncertainties);
- **independent** (random, globally uncorrelated);
- **structured** (correlated along defined distances, **with correlation length and function.**

Each component is propagated separately, and at the end of the L2A processing, all contributions with the same correlation structure (i.e., all common, all independent, all structured) are summed up (according to the GUM [4], as the square root of the squared contributions) according to Equation (4)—this summing up of uncertainty contributions with different correlation structures has to be performed at each aggregation level after having propagated each contributions separately according to its correlation structures.

$$u(\text{AOD}) = \sqrt{\left(\frac{\partial \text{AOD}}{\partial R_{\text{TOA}}} u\left(R_{\text{TOA}}^{\text{independent}}\right)\right)^2 + \left(\frac{\partial \text{AOD}}{\partial R_{TOA}} u\left(R_{\text{TOA}}^{\text{structured}}\right)\right)^2 + \left(\frac{\partial \text{AOD}}{\partial R_{TOA}} u\left(R_{\text{TOA}}^{\text{common}}\right)\right)^2} \qquad (4)$$

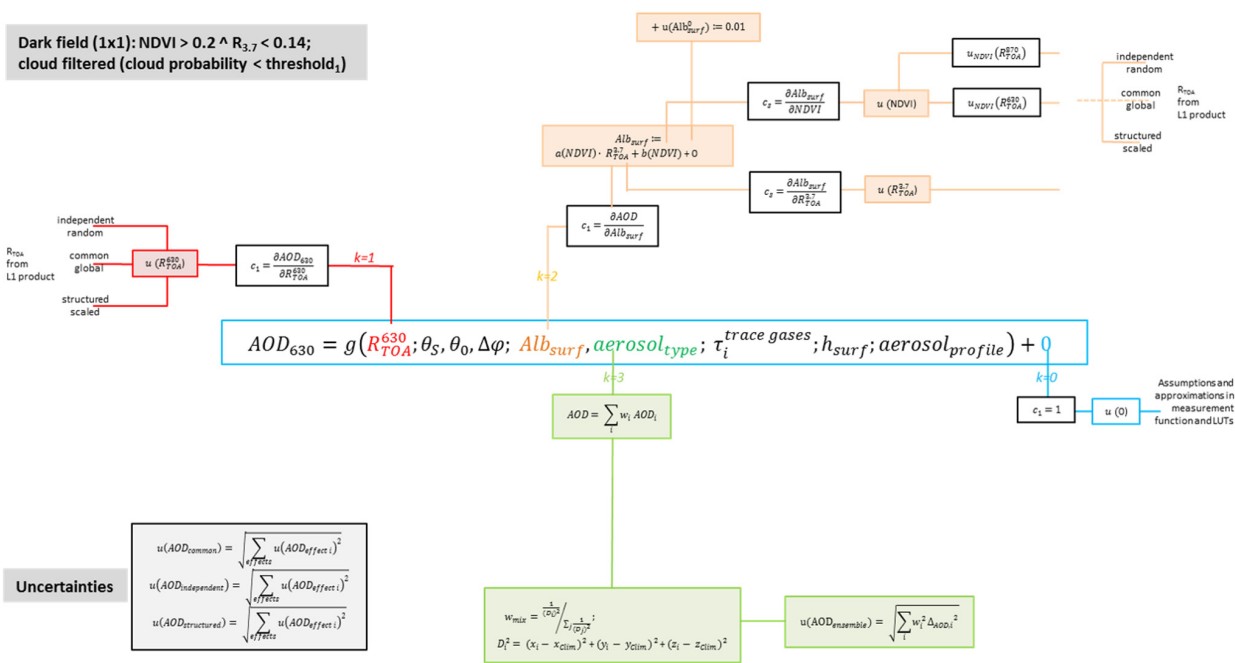

**Figure 2.** L2A (dark field inversion) measurement-function centered diagram.

## 2.4. Characterization of the Major Uncertainty Contributions

According to the FIDUCEO recipe, each uncertainty contribution in Equation (3) needs to be characterized as well as possible regarding the maturity of understanding and quantifying it at the measurement (=pixel) level and regarding its correlation structures in space and time. To systematize this analysis and cover both quantitative information (where detailed studies have been conducted) as well as qualitative initial assessments (where no detailed analysis has yet been made), the FIDUCEO recipe has defined "effects tables", as will be discussed in the following sub-sections. Note that the following tables apply the same colour coding as in Equations (2) and (3) and in Figures 1 and 2.

### 2.4.1. Direct Reflectance Inversion (L2A, Process 1)

The direct radiance inversion uses a pre-calculated look-up table for one aerosol type (specific choice of optical properties) to invert a top-of-atmosphere reflectance in the red band into an AOD value. The uncertainties of this process are characterised as shown in Table 1, the main source of uncertainty being the propagated reflectance uncertainties in the red channel (taken from the level 1 input product) with sensitivities defined by partial derivatives of the LUT vs. reflectance.

The sensitivity of the AOD to the measurement quantity $R_{\text{TOA}}$ depends on several parameters (e.g., geometry). While scattering by aerosols increases the top-of-atmosphere reflectance with growing AOD, extinction of the reflection from the underlying surface by aerosols also decreases the top-of-atmosphere reflectance with growing AOD. By the counter-play of these two effects, the sensitivity depends on the surface brightness and can become very low or even zero at a certain medium value of $Alb_{\text{surf}}$. Such conditions with weak sensitivity of $R_{\text{TOA}}$ on AOD lead to very large uncertainties and must be avoided. This is the main purpose of using a dark-field approach (for dark surfaces with $Alb_{\text{surf}}$ close to zero, the sensitivity is the highest and the respective AOD uncertainty is the lowest).

**Table 1.** Effects table for direct reflectance inversion.

| Table Descriptor (k = 1) | | Quantity | Notes |
|---|---|---|---|
| Name of effect | | Direct reflectance inversion | |
| Affected term in measurement function | | AOD = $f(R_{TOA})$ | |
| Maturity of analysis | Maturity of uncertainty estimate | 3—rigorous analysis | Online analysis from partial derivatives along one albedo line of look-up tables (LUT) and propagated L1B uncertainties |
| | Maturity of correlation scale estimate | 3—Strong evidence (L1B) | |
| | If maturity of estimate is 0 or 1, how significant do you expect this effect to be? | significant | |
| Correlation type and form | From level 1 | As in FIDUCEO easyFCDR | Easy FCDR: separated in 2 bulk contributions (unstructured random and globally structured) |
| | Larger scale temporal [time] | | |
| | Larger scale spatial [geospatial coordinates] | | |
| Correlation scale | From level 1 | As in FIDUCEO easyFCDR | |
| | Larger scale temporal [time] | | |
| | Larger scale spatial [geospatial coordinates] | | |
| Uncertainty | PDF shape | from FIDUCEO easyFCDR | |
| | units | Units of $R_{TOA}$ | |
| | magnitude | From FIDUCEO easyFCDR $R_{TOA}$ | |
| Sensitivity coefficient | | $c_1 = \dfrac{\partial AOD_{670}}{\partial R_{TOA}^{670}}$ | Partial derivatives of LUT |

## 2.4.2. Albedo Dependence in Inversion (L2A, Process 2)

The albedo dependence in the inversion leads to selecting two suitable lines (interpolating between them) in a pre-calculated look-up table for one aerosol type (specific choice of optical properties) to invert a top-of-atmosphere reflectance in the red band into an AOD value for the appropriate surface albedo. The uncertainties of this process are characterised as shown in Table 2, the main source of uncertainty being propagated reflectance uncertainties (NDVI and mid-infrared channel) with sensitivities defined by partial derivatives of the LUT between two polynomials of adjacent surface albedo values. It should be noted that in our analysis, no cross-band correlations are considered, which could impact the NDVI uncertainties (calculated from red and near-infrared reflectances).

## 2.4.3. Aerosol Type Ensemble Uncertainty (L2A, Process 3)

The inversion depends on a combination of several optical aerosol properties ($\sigma_e$ ($\lambda$), $\omega_0$, $p(\psi)$ from Equation (1), together named aerosol type). The choice of aerosol type determines which pre-calculated look-up table is selected to invert a top-of-atmosphere reflectance in the red band into an AOD value for the appropriate surface albedo. The uncertainties associated with the choice are characterized in Table 3. In this case, no separation into input uncertainty and sensitivity is made, but an ensemble approach is applied, where the inversion is repeated 36 times for a set of pre-defined aerosol types (aimed to cover all realistic mixing possibilities in the atmosphere) and a model-based mixing fraction climatology is used as the auxiliary dataset to prescribe the most likely aerosol type, while the spread (weighted with closeness to this selected aerosol type) gives an indication of the uncertainty due to lacking knowledge of the aerosol type.

**Table 2.** Effects table for the albedo dependence in the inversion.

| Table Descriptor (k = 2) | | Quantity | | Notes |
|---|---|---|---|---|
| **Name of effect** | | Albedo dependence in inversion | | |
| **Affected term in measurement function** | | $AOD = f(Alb_{surf})$ | | |
| | | $Alb_{surf} = f(NDVI, R_{1.6})$ | parameterization | |
| **Maturity of analysis** | Maturity of uncertainty estimate | 2—Some analysis performed to estimate values | | Online analysis from partial derivatives between two albedo lines of look-up tables (LUT) and estimated surface albedo uncertainties |
| | Maturity of correlation scale estimate | 2—Based on analysis, unsure about correlation shape ($Alb_{surf}$) | | |
| | If maturity of estimate is 0 or 1, how significant do you expect this effect to be? | significant | | Albedo estimated with NDVI and $R_{1.6}$ |
| **Correlation type and form** | From level 1 | As in FIDUCEO easyFCDR | rectangle_ absolute | |
| | Larger scale temporal [time] | | | |
| | Larger scale spatial [geospatial coordinates] | | | |
| **Correlation scale** | From level 1 | As in FIDUCEO easyFCDR | | |
| | Larger scale temporal [time] | | global | |
| | Larger scale spatial [geospatial coordinates] | | global | |
| **Uncertainty** | PDF shape | from FIDUCEO easyFCDR | rectangle | |
| | units | Units of $Alb_{surf}$ | | |
| | magnitude | Random: $\partial Alb_{surf} = \sqrt{\left(\frac{\partial Alb_{surf}}{\partial NDVI} \cdot u(NDVI)\right)^2 + \left(\frac{\partial Alb_{surf}}{\partial R_{TOA}^{1.6}} \cdot u(R_{TOA}^{1.6})\right)^2}$ | 0.01 | $u$ (NDVI) and $u$ ($R_{1.6}$) propagated from FIDUCEO easyFCDR $R_{670}$, $R_{870}$, and $R_{1.6}$ |
| **Sensitivity coefficient** | | $c_1 = \frac{\partial AOD}{\partial Alb_{surf}}$ | | Partial derivatives of LUT |

**Table 3.** Effects table for the aerosol type dependence in the inversion.

| Table Descriptor (k = 3) | | Quantity | Notes |
|---|---|---|---|
| **Name of effect** | | Aerosol type ensemble | |
| **Affected term in measurement function** | | $AOD = f(aerosol\ type)$ | |
| **Maturity of analysis** | Maturity of uncertainty estimate | 2—Some analysis performed | Aerosol type determines choice of appropriate LUT |
| | Maturity of correlation scale estimate | 1—Estimated | Estimated with an ensemble of AOD solutions encompassing pre-defined aerosol type set weighted by distance in mixing fraction space $x_i, y_i, z_i$ to a climatology most probable type (per month and 1 deg lat-lon); climatology contains best knowledge median from ~10 AEROCOM aerosol models |
| | If maturity of estimate is 0 or 1, how significant do you expect this effect to be? | significant | |
| **Correlation type and form** | From level 1 | | Uncertainty of model-based mixing fraction climatology is not quantified |
| | Larger scale temporal [time] | Rectangle_absolute | Uncertainty due to spread of AOD solutions with different aerosol types is rigorously calculated with the ensemble |
| | Larger scale spatial [geospatial coordinates] | Rectangle_absolute | |
| | From level 1 | | |
| | Larger scale temporal [time] | 1 week | Scales of the grid of the aerosol type climatology |
| | Larger scale spatial [geospatial coordinates] | 1 degree | |
| **Uncertainty** | PDF shape | rectangle | |
| | units | Units of AOD | |
| | magnitude | $u(AOD_{ensemble}) = \sqrt{\sum_i w_i^2\, \Delta_{AOD,i}^2}$ | $\Delta AOD_i = AOD_i - AOD;$ $w_{mix} = \frac{\frac{1}{(D_i)^2}}{\Sigma_j \frac{1}{(D_j)^2}};$ $D_i^2 = (x_i - x_{Clim})^2 + (y_i - y_{Clim})^2 + (z_i - z_{Clim})^2$ |
| **Sensitivity coefficient** | | 1 | uncertainty and sensitivity coefficient cannot be separated |

In order to estimate an appropriate error correlation length scale, the question needs to be answered how the error (the difference between the assumption made about aerosol type and the true aerosol type) is common from one pixel/super-pixel to the next. The error correlation length scale is then the scale on which the difference between the assumed

aerosol type and the unknown true aerosol type is highly consistent. Firstly, for the error correlation scale, we consider that of the underlying climatology grid. In the spatial dimension, aerosol type is common over relatively large areas, and the dominant error correlation scale is that of the climatology grid, namely 1°. Temporally, the climatology is provided as monthly averages. In practice, the aerosol type will typically change over smaller timescales of ~1 week. Therefore, an error correlation scale of 1 week is used.

### 2.4.4. Cloud Mask-Induced Uncertainty (L2B, Process 4)

Assessing uncertainties due to uncertainties in the cloud masking is performed by repeating the super-pixel aggregation with a set of dark fields obtained under two different values of the cloud probability threshold. The difference of those two super-pixel aggregates is used as estimate of the cloud-mask-induced AOD uncertainty. In spatially homogeneous conditions, this uncertainty will be close to 0, while it is expected to show larger values for inhomogeneous conditions (broken clouds, edge of large cloud area, but also high optical depth aerosol plumes). The uncertainties of this process are characterized as shown in Table 4.

**Table 4.** Effects table for cloud-mask induced uncertainty.

| Table Descriptor (k = 4) | | Quantity | Notes |
|---|---|---|---|
| Name of effect | | Cloud mask uncertainty induced AOD uncertainty | |
| Affected term in measurement function | | | Can only be estimated on L2B superpixel level ($10 \times 10$ km$^2$) |
| Maturity of analysis | Maturity of uncertainty estimate | 1—Rough estimates only | Is estimated by using 2 different thresholds for cloud probability and then calculating mean AOD with remaining selected pixels |
| | Maturity of correlation scale estimate | 1—Estimated | |
| | If maturity of estimate is 0 or 1, how significant do you expect this effect to be? | significant | Setting of the two thresholds needs to be optimized |
| Correlation type and form | From level 1 | | Clouds are changing extremely fast |
| | Larger scale temporal [time] | Random | |
| | Larger scale spatial [geospatial coordinates] | random | |
| Correlation scale | From level 1 | | Clouds are changing extremely fast |
| | Larger scale temporal [time] | - | |
| | Larger scale spatial [geospatial coordinates] | - | |
| Uncertainty | PDF shape | Random (temporal) random (spatial) | |
| | units | Units of AOD | |
| | magnitude | $u(AOD_{cloud\ mask}) = AOD_{3\times3}^{mean}(threshold_1) - AOD_{3\times3}^{mean}(threshold_2)$ | |
| Sensitivity coefficient | | 1 | |

### 2.4.5. Look-Up Table Noise

Look-up tables have been obtained with high accuracy radiative transfer forward calculations for a large set of parameters (AOD, surface albedo, geometries, aerosol types) and then fitting second-order polynomials to them. The steps in the discretization have been chosen to keep the errors of the fitting curves small against the exact calculations—here, we quantify an average uncertainty of 0.01 AOD for this small LUT interpolation uncertainty as elaborated in Table 5. The underlying radiative transfer calculations are regarded as error-free (or at least one order of magnitude smaller).

**Table 5.** Effects table for LUT interpolation.

| Table Descriptor (k = 0) | | Quantity | Notes |
|---|---|---|---|
| Name of effect | | Look-up table noise | |
| Affected term in measurement function | | AOD = f($R_{670}$; Alb$_{surf}$, aerosol type) | |
| Maturity of analysis | Maturity of uncertainty estimate | 2—Some analysis performed to estimate values | One LUT for each aerosol type |
| | Maturity of correlation scale estimate | 1—Estimated | |
| | If maturity of estimate is 0 or 1, how significant do you expect this effect to be? | Minor | |
| Correlation type and form | From level 1 | | |
| | Larger scale temporal [time] | Exponential_decay | |
| | Larger scale spatial [geospatial coordinates] | Exponential_decay | |
| Correlation scale | From level 1 | | |
| | Larger scale temporal [time] | 5 days | |
| | Larger scale spatial [geospatial coordinates] | 100 km | Typical aerosol lifetime / plume size—correlated within LUT |
| Uncertainty | PDF shape | exponential | |
| | units | Units of AOD | |
| | magnitude | 0.01 | |
| Sensitivity coefficient | | 1 | |

### 2.4.6. Minor Uncertainty Contributions

Analysis of the literature and radiative transfer simulations have been used to identify the major sources of uncertainties in passive aerosol remote-sensing, which are treated in the uncertainty propagation discussed in the earlier sections of this chapter. Table 6 lists other potential sources of uncertainty which are regarded as minor (included into the second-order uncertainty term $u^2(0)$) with a short description and a quantitative estimate of their relevance at pixel and grid/monthly average level.

**Table 6.** Effects which are regarded as minor or negligible.

| Source of Uncertainty (Measurement Function Term Affected, if Appropriate) | Description | Likely Sensitivity of Output to This | |
|---|---|---|---|
| | | On Small Scales | On Large Scales |
| Vertical aerosol profile $\sigma_e$ (z) | Different assumptions are made for different aerosol types but sensitivity at TOA is small for VIS/IR sensors | low for mid-visible bands | low for mid-visible bands |
| Directional reflectance ratio $\gamma$surf ($\theta_s$, $\theta_0$, $\Delta_\varphi$) | Directionality of surface reflectance (treated by estimating surface albedo from mid-infrared signal which is in first order exhibiting the same directionality) | Medium, but difficult to quantify from nadir only observations | Low-medium, since it averages out by averaging different surface types |
| Trace gas concentration profiles and absorbing cross sections (from laboratory) $\tau_i$ | Critical absorption bands are usually avoided so that total band absorption even for high concentration/low angles is on the order of few percent (and its uncertainty mostly below 1%) | Low | Low-medium in case of long-term trends of trace gas concentrations |
| Radiative transfer forward model | Typical accuracy of simulated reflectance < 1% (and thus smaller than propagated L1B uncertainties) | Low | Low |
| Overpass time | Polar orbiting sensors provide typically one or two sun-synchronous overpass times per day | High when linking sensors of different platforms, when there is a significant time shift | High when linking sensors of different platforms, when there is a significant time shift |
| altitude h$_{surf}$ dependent Rayleigh scattering correction | The small reflectance due to molecular scattering is reduced with increasing altitude | Low in red band | Low in red band |

**Table 6.** *Cont.*

| Source of Uncertainty (Measurement Function Term Affected, if Appropriate) | Description | Likely Sensitivity of Output to This | |
|---|---|---|---|
| | | On Small Scales | On Large Scales |
| Response function uncertainties | Change trace gas absorption and effective wavelength used for Rayleigh and for aerosol radiative transfer calculations | Low for red band where radiative transfer calculations are made | Low for red band where radiative transfer calculations are made |
| AOD conversion 630 -> 550 nm | Is accurately determined once aerosol type is specified; remaining uncertainties by discretization of aerosol types (interpolation between them) and by definition of aerosol components (specified to encompass natural variability) | Low, since uncertainty of aerosol type ensemble is assessed | Low, since uncertainty of aerosol type ensemble is assessed |

### 2.5. Uncertainty Propagation to Higher CDR Product Levels

The propagation of uncertainties from L2A to L2B (and similar to L3) is then determined by the **FIDUCEO analysis tree** (L2B, Figure 3). Here, the correlation structures are now taken into account. The independent contributions (no correlation at all) can be squared (Equation (5a)); thus, this noise term is reduced by $1/\sqrt{N}$ with an increasing number of pixels N. In contradiction, the common contributions are simply averaged (Equation (5b)) and achieve no reduction with growing number N. In between those two extremes, the structured contributions (Equation (5c)) depend on the correlation $c_{ij}$ (which typically decrease with growing distance of elements I and j). In the end, the total super-pixel uncertainty is then summed up from these three parts with Equation (4) (squared, as they are independent from each other):

$$u\left(\text{AOD}_{3\times3}^{independent}\right) = \frac{1}{N_{dark\ fields}} \cdot \sqrt{\sum_{dark\ fields} u\left(\text{AOD}_{dark\ field\ i}\right)^2} \quad (5a)$$

$$u\left(\text{AOD}_{3\times3}^{common}\right) = \frac{1}{N_{dark\ fields}} \cdot \sum_{dark\ fields} u\left(\text{AOD}_{dark\ field\ i}\right) \quad (5b)$$

$$u\left(\text{AOD}_{3\times3}^{structured}\right) = \frac{1}{N_{dark\ fields}} \cdot \sqrt{\sum_{dark\ fields} u\left(\text{AOD}_{dark\ field\ i}\right)^2 + 2\sum_{i>j}\sum c_{ij}\, u\left(\text{AOD}_{dark\ field\ i}\right) u\left(\text{AOD}_{dark\ field\ j}\right)} \quad (5c)$$

One particular element of the super-pixel uncertainty is the contribution due to uncertainties of the cloud masking. This effect can only be estimated from the L2B super-pixel level, as shown in Figure 3. The cloud retrieval algorithm APOLLO_NG [14] used results in a Bayesian cloud probability, so we can derive two different cloud masks (weak and strong) by defining two different probability thresholds (5% and 50%). The AOD retrieval is then used for all cloud-free pixels of either cloud mask, and the average AOD per super-pixel cell is calculated. The AOD difference between conservative and relaxed cloud-masking is then used as a proxy for the cloud-mask-induced uncertainty.

The gridded L3 AOD values are then determined with a simple aggregation of all super-pixel values within the grid cell, as shown in Figure 4. We provide daily and monthly 1-degree latitude/longitude gridded data.

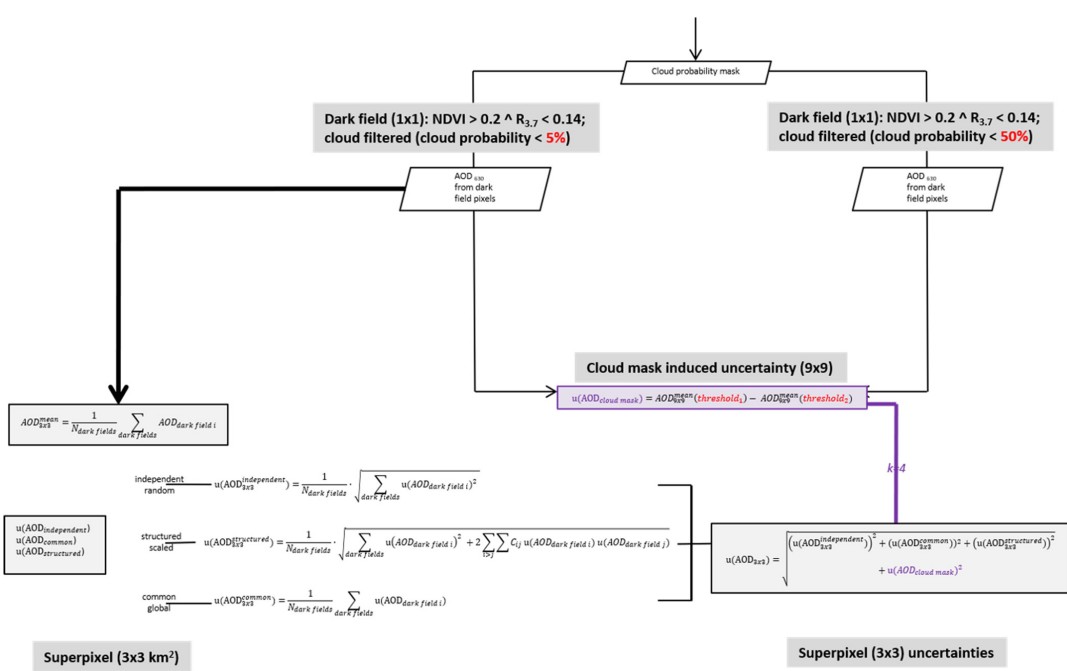

**Figure 3.** L2B analysis tree for AVHRR AOD.

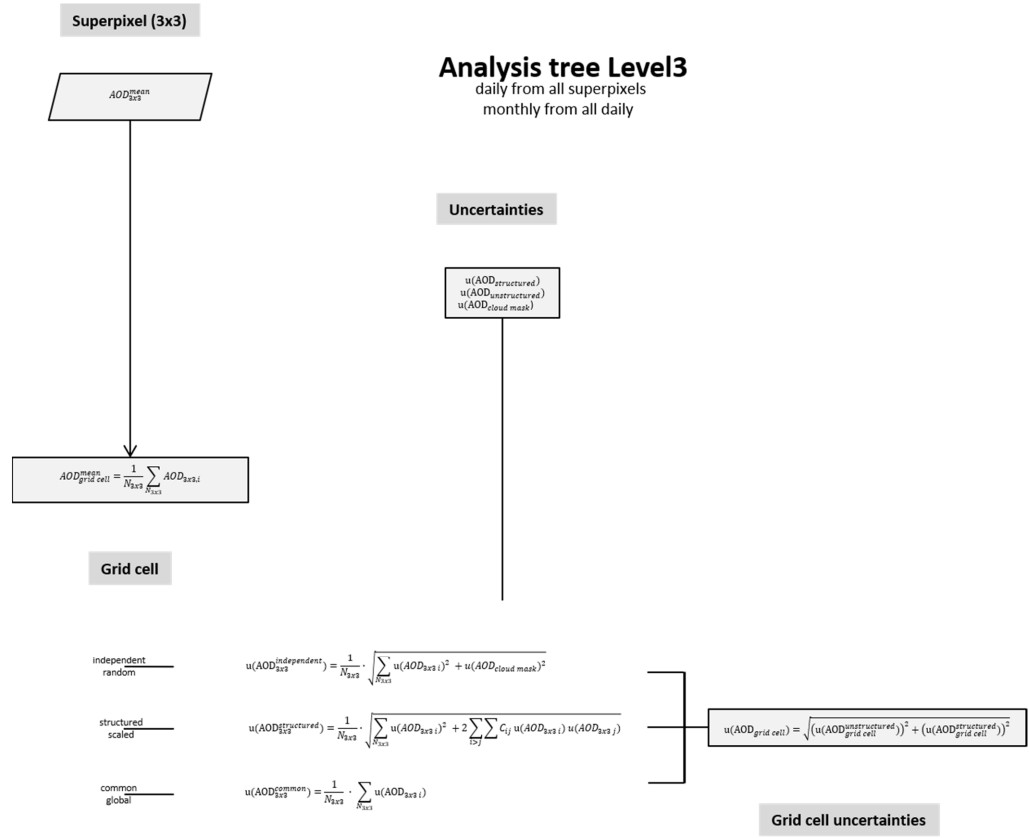

**Figure 4.** L3 analysis tree for AVHRR AOD.

In propagating the structured contributions, the correlation functions of the errors (their probability distribution form and their length scale) in space and time dimension need to be known or estimated for each effect—note that this is not the correlation of the physical quantities but the correlation of their errors. The choices identified according to the FIDUCEO approach are shown in Table 7. Note that this information is directly used from the L1B product for the reflectance and surface albedo effects, while it needs to be estimated from physical understanding of aerosol plumes and cloud systems.

The AOD for each super-pixel and grid cell is determined as described in Figures 3 and 4. Aggregated AOD is calculated as the mean of AOD values of all (dark field) pixels. In the averaging, no weighting with dark-field uncertainties is conducted since these uncertainties depend strongly on AOD values themselves and could therefore introduce a bias to a subfraction of retrieved AOD values (either high or low, depending on the geometry, surface brightness, and albedo). In addition, at this processing level, the uncertainty induced by uncertainties in the cloud masking is estimated from the AOD difference obtained with two different thresholds of cloud probability. Due to the simplicity of these equations, a measurement-function-centred diagram is not provided for this step. The sources of uncertainty are given in Table 8.

**Table 7.** Averaging the different effects to super-pixels and gridded products.

| Effect | Uncertainty Correlation Structure | Spatial Correlation Daily Gridded Data | Temporal Correlation Monthly Gridded Data |
|---|---|---|---|
| TOA reflectance | Common within line Structured across lines Uncorrelated in time | pdf from FIDUCEO easyFCDR Level1b ($R_{0.63}$) | - |
| Surface albedo | Common within line independent across lines Uncorrelated in time | pdf from FIDUCEO easyFCDR Level1b ($R_{3.7}$, NDVI) | - |
| Aerosol type | Climatology grid Typical aerosol lifetime | 1°/rectangular | 1 week/rectangular |
| Cloud mask | None (extremely short cloud lifetime) | - | - |

**Table 8.** Sources of uncertainty.

| Measurement Function Term | Source of Uncertainty | Sensitivity Coefficient | Comment |
|---|---|---|---|
| $AOD_{3\times3}$ | Propagated uncertainty from L2B, separate for independent, structured and common contributions | $\frac{1}{N}$ | There are $N$ such terms |
| $+0$ | Representativeness of measured pixels within the area | 1 | |

## 3. Evaluation of Propagated AVHRR AOD Uncertainties

For a demonstration of uncertainty propagation from the easyFCDR L1B input, a 10-year AOD record over Europe and North Africa was processed from two subsequent AVHRR/3 instruments onboard NOAA-16 and NOAA-18, as shown in Table 9. The geographic coverage of the record consists only of pixels contained in the orbit L1B files over land within the rectangular area between latitudes of 30 and 75 degrees north and longitudes of 10 degrees west and 75 degrees east (see Figure 5).

**Table 9.** Temporal coverage and satellites that are included in this AVHRR aerosol demonstration CDR.

| Satellite | Initial Equator Crossing Time | Period Processed |
|---|---|---|
| NOAA-16 | 14:30 | 1 January 2003–31 December 2005 |
| NOAA-18 | 13:30 | 6 June 2005–31 December 2012 |

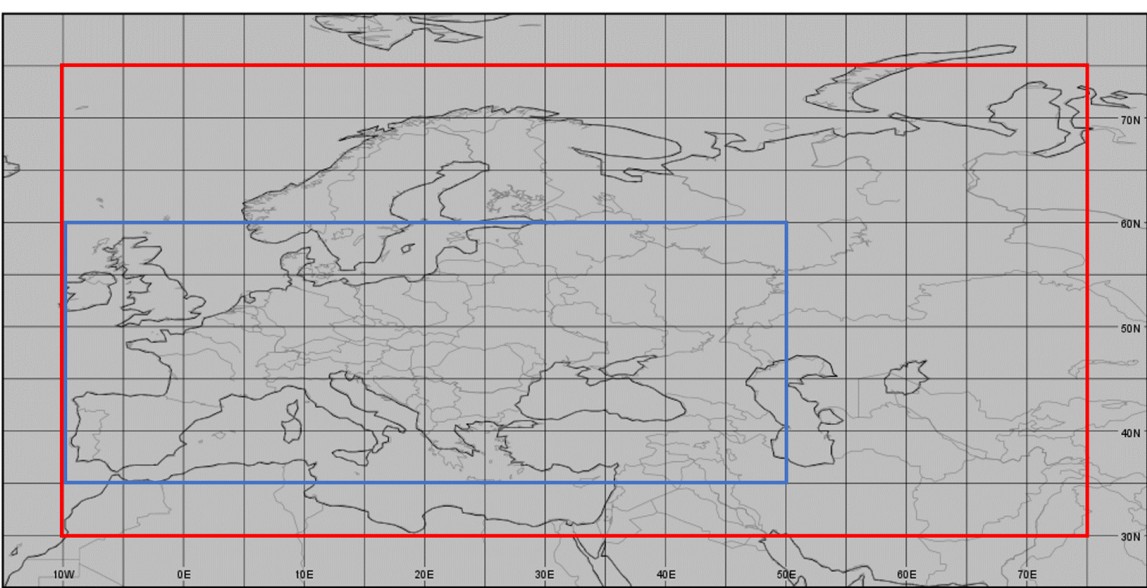

**Figure 5.** Regions covered by the dataset in this paper: "Europe and North Africa", targeted by this study (in red, Section 3.1). "Europe" for comparison with [16] (in blue, Section 3.2).

### 3.1. Evaluation of the Propagated AOD Uncertainties

For the validation of the FIDUCEO AVHRR AOD uncertainties, we use the common standard in the aerosol retrieval community, which is comparison to ground-based sun photometer measurements. These can directly measure exactly the same quantity, namely the aerosol optical depth (AOD), with very high accuracy (~0.01) by directly looking at the sun. A continuous network is coordinated by NASA (AERONET [17]), from which the latest processed version v3 of quality-controlled sun photometer measurements at ~200 permanent stations back to the mid-1990s can be obtained. This provides a unique reference dataset. However, even this unique reference has its limitations in global coverage (less so in Europe and the USA), and for larger pixels or grid cells, there is a representativity issue of a point station measurement against a large area covered from satellite—this becomes a significant limitation for 1-degree latitude–longitude grid cells (imagine how different environmental conditions can be within a rectangular area of ~110 km size). We statistically analyse matches between AVHRR and AERONET (within the commonly used matching window of ±30 min and ±50 km from a pixel/grid cell center).

As a first assessment of how far the uncertainties match the true errors, the ratio of error (AVHRR AOD–AERONET AOD) over uncertainties is calculated (named "normalized error") after [7] and probability distributions of the normalized error are plotted as shown in Figure 6. We plot the probability distributions and the cumulative distributions in comparison to a Gaussian distribution with the same standard deviation (0.145). This analysis shows a reasonable agreement of the real AVHRR uncertainty distributions with the theoretical Gaussian distributions (except for larger normalized errors in the long tails) and proves a reasonable statistical agreement of uncertainties and errors.

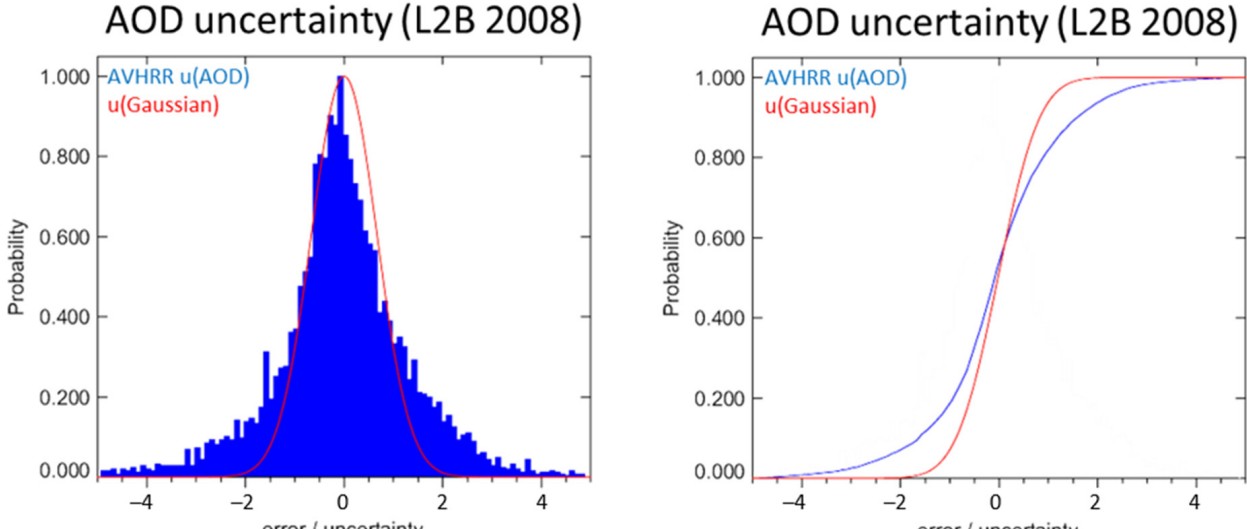

**Figure 6.** Probability distributions of the normalized error compared in the L2B product for 2008 to a Gaussian distribution of the same width: distribution (**left**) and cumulative distribution (**right**); the AVHRR AOD uncertainty histograms are plotted in blue, and the theoretical Gaussian distribution of identical width in normalized errors (1.5) are plotted in red.

For a quantitative analysis, we developed a new approach to assess the uncertainties by comparing probability functions of the difference between AVHRR and AERONET AOD (a very good estimate of the true error) with the distributions of errors derived from the propagated uncertainties through the processing chains levels (L2A, L2B, L3_daily). It should be noted that an uncertainty is not one concrete realization of an error but represents the standard width of an error distribution. Therefore, by super-imposing those (normalized) Gaussian error distributions of each measurement with a width given by its pixel uncertainty, we obtain these derived error histograms from the uncertainties. Figure 7 shows a good agreement of the true error distribution (standard deviation of 0.16) with the uncertainty-derived distribution, where the error distribution shows a small negative bias (standard deviation of 0.164). For illustration, we also compare with two hypothetic distributions which were achieved by propagating all uncertainties, assuming either their full correlation between pixels (all common, standard deviation of 0.18) or full independence (all random, standard deviation of 0.10), leading to a wider/much smaller distribution.

In order to assess the suitability of uncertainties to differentiate pixels with likely small from likely large errors, in Figure 8, we plot three percentile values (38%, 68%, 95%) of the absolute error distributions for each (binned) value of the uncertainties (following [8])—this is representative of plotting the full error histograms along the *y*-axis along each value of the associated propagated uncertainty at the *x*-axis. These three percentage values should lie along linear functions with gradients of 0.5, 1, and 2. Most points with u(AOD) between 0.05 and 0.25 lie close to the expected 1:0.5 and 1:1 lines, which indicates that their uncertainties are a meaningful predictor for the expected size of the errors. However, the points at the edges (left and right, and upper) are further away from them. The latter fact can be explained by low numbers in bins with very low (u(AOD) < 0.05) and very high (u(AOD) > 0.25 uncertainty, as shown in Figure 7, and for normalized errors above 2 (the blue symbols in Figure 8); all these lead to high binning errors in the percentile values.

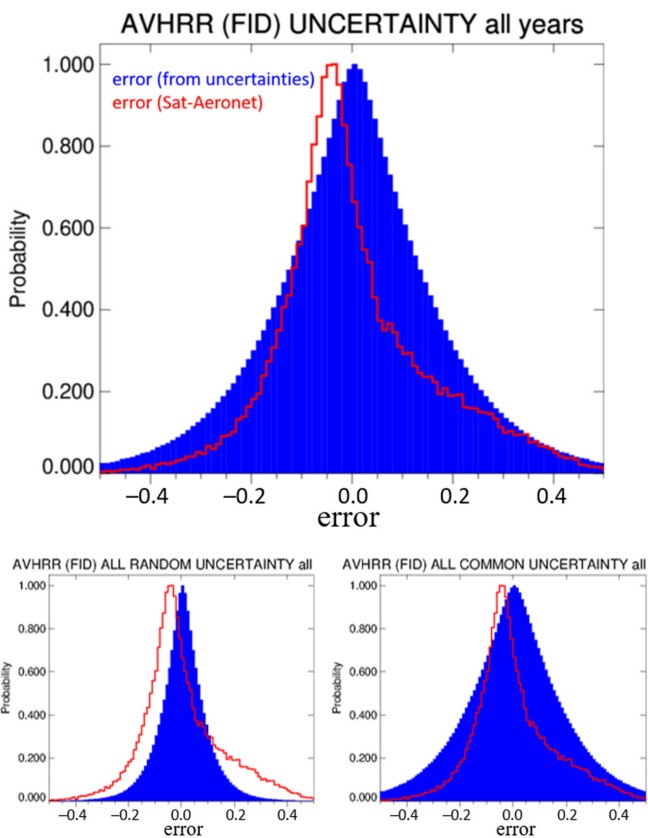

**Figure 7.** Probability distributions of the estimated true error vs. AERONET (in red) and errors derived from uncertainties (in blue) for the 10 year AVHRR AOD dataset. Top: distribution for propagated uncertainties; bottom: hypothetic distributions for "all random" (**left**) and "all common" (**right**) propagated uncertainties.

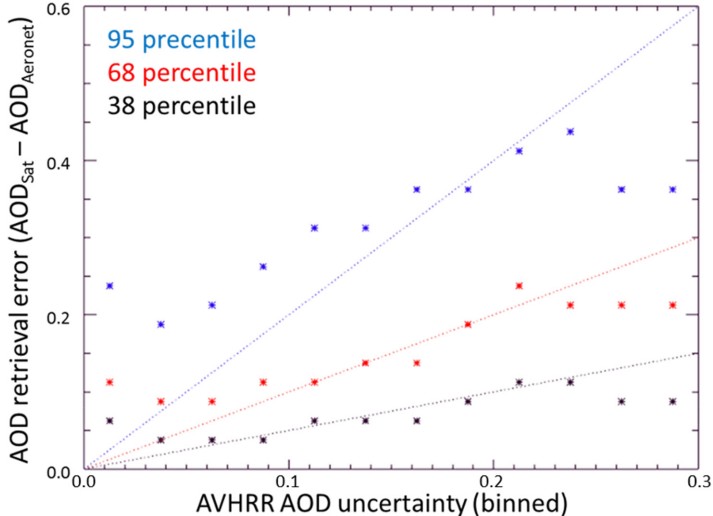

**Figure 8.** Assessment of the information content of uncertainties in the L2B product for 2008: 3 percentile values (for 38%, 68%, and 95% in black, red, and blue) of the retrieval error as function of the predicted uncertainty; expected functionality is depicted with the dashed lines of gradient 0.5, 1.0, and 2.0.

Finally, in Figure 9, we assess the distributions of propagated uncertainties for 2008 for different product levels: L2A (single pixels, standard deviation of 0.16), L2B (super-pixels

from 3 × 3 single pixels, standard deviation of 0.145), and L3 daily (1-degree, standard deviation of 0.12) versus their true error distributions (in all cases using the same matching criteria and thus neglecting the effects of sampling/representativity). All three plots show a good agreement of the true error and uncertainty-derived distributions (with a small negative bias for the true error histograms), which indicates that the propagation to higher product levels provides meaningful uncertainties.

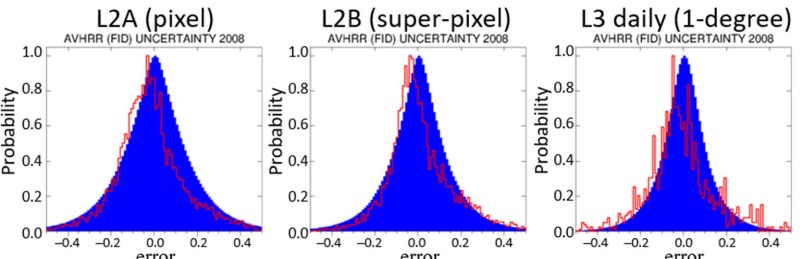

**Figure 9.** Probability distributions of the estimated true error vs. AERONET (in red) and errors derived from uncertainties (in blue) for different product levels of the AVHRR AOD dataset (2008): (from **left** to **right**) pixel level, super-pixel level, daily gridded level.

### 3.2. Comparison of the AOD CDR and Its Uncertainties to a Reference CDR

We show the monthly mean time series aggregated for the whole of Europe (over land only, 10W–50E, 35–60 N, excluding parts of Algeria and Morocco in this rectangle) and its propagated AOD uncertainty in Figure 10. We compare with a multi-sensor merged dataset exactly for the same area including the period 2003–2012 [16]. Figure 10 shows the good agreement (with lower minima for AVHRR, possibly due to different coverage), and also good agreement in the seasonal cycle; even a double peak in summer is visible, while the intra-annual maxima/minima variation disagrees (likely due to the different sampling). We can also see that the overlap of the AVHRR record parts from the two platforms NOAA-16 (black) and NOAA-18 (red) in 2005 (7 months) in Figure 10 is very good. Finally, the agreement between the merged and the AVHRR AOD records lies largely within the AVHRR propagated AOD uncertainty range.

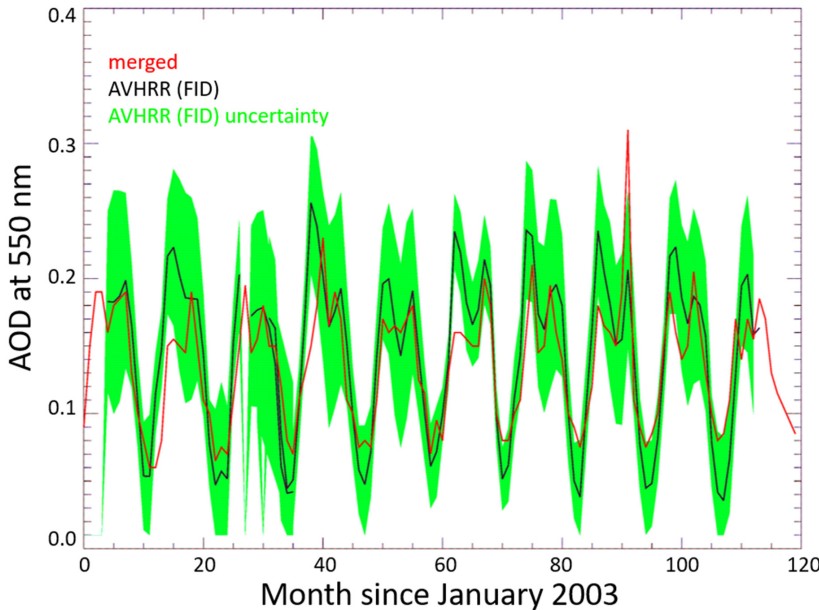

**Figure 10.** Monthly AOD records over Europe (land only): FIDUCEO AVHRR and multi-sensor merged dataset [16].

## 4. Discussion

This paper provides a demonstration dataset of the aerosol optical depth (AOD) over land over Europe and North Africa inferred from the AVHRR instrument to illustrate the propagation of uncertainties benefitting from the methodology and the easyFCDR L1B dataset, which were both developed in the Horizon2020 project FIDUCEO. In terms of AOD retrieval, this demonstration dataset, which can only exploit one single channel over land, is, as expected, comparatively weak (against other more sophisticated sensors with many more observables usable for aerosol properties). However, despite the very large scatter of the pixel results, we can show good skill of a 10-year climate data record for Europe which agrees well in its seasonality and patterns with a community-merged data record within the ranges of the propagated AVHHR AOD uncertainties.

The significant strength of this demonstration dataset is that its uncertainties could be propagated taking into account detailed correlation structures. This was enabled by using the FIDUCEO easyFCDR L1B AVHRR dataset as the input, which provides uncertainties separated into three bulk contributions: independent (uncorrelated, random), structured (regionally correlated, e.g., within one sensor line or column, with a correlation function along the distance), and common (globally correlated). Consequently, the uncertainties on averaged product levels (L2B and L3 daily) see some reduction due to averaging, but significantly less as if all uncertainties were considered random. This is due to the fact that within the visible bands of the AVHRR FCDR product, the common (globally correlated) contribution dominates, and also the uncertainties introduced by the surface albedo estimation as the most difficult part of the retrieval are non-zero and significant.

The FIDUCEO systematic methodology to analyse the processing chain and the propagation of uncertainties through it proved very useful to structure this analysis including the direct propagation of uncertainties from the input L1B products and also the estimation of uncertainties from other dominant sources within the measurement equation for which auxiliary information is used. The FIDUCEO methodology provides guidance to at least estimate (if not enough time is available to conduct comprehensive sensitivity studies) all needed properties for uncertainty propagation for the dominant effects and their correlation structures in space and time.

The uncertainties in the FIDUCEO L1b AVHRR reflectances/brightness temperature easyFCDR product which contain all uncertainties grouped into three contributions with different correlation structures (uncorrelated "independent", globally correlated "common" and regionally/periodically correlated "structured") are able to manage uncertainty propagation through the different processing levels. This practical input uncertainty information is appropriate for downstream usage—the fullFCDR product contains many different detailed uncertainty contributions which would be much more demanding for a user to handle. The L1B uncertainty information provided was shown to be suitable to model, during spatial/temporal aggregation, the right balance between uncertainties that exhibit a noise reduction and uncertainties that are pertained.

We validate the distributions of propagated uncertainties at different product levels against best estimates of the true error (difference with the almost error-free AERONET measurements) to prove good agreement and a reduction in uncertainties through the processing chains levels (L2A, L2B, L3_daily) by averaging.

The use of the CDR uncertainties can be directly derived from the intrinsic use of the uncertainties during the propagation of the different processing levels where in each application a user needs to take similar steps. On all levels of the AVHRR AOD product, the user receives total uncertainties (which can be used directly, for example, in a data-assimilation scheme or to ascertain a confident range around values). However, the contributions following up from the input L1B component are also provided, which allow a user to take their different correlation structures into account when averaging spatially or temporally. The independent contribution is fully random and can be added up by calculating the square root of the sum of squared pixel uncertainties divided over the number of pixels (this leads to the noise reduction increasing with the number of averaged pixels or grid cells). The

common contribution is always assumed to be globally correlated so that uncertainties from different pixels can be simply averaged (no noise reduction can thus be achieved). For propagation of the structured contributions, their inherited correlations from the input L1B data need to be applied. For the gridded datasets, all structured contributions lose their correlations and are treated as independent, with one exception: the aerosol-type ensemble uncertainty is fully correlated within a 1-degree grid cell and within a week.

We have also analysed the feasibility of using uncertainty to select best pixels (as was requested at AEROSAT/AEROCOM meetings by model users). The key issue is whether a filtering with low uncertainty entails the risk of suppressing high AOD values. As part of the product files, we provide a variable AODBEST filtered with AOD uncertainty <0.15 for the gridded processing levels (L3 daily, monthly). This filtering typically keeps the key AOD features (slightly reducing parts) but somewhat reduces the coverage (not too critically) and erases highly uncertain data.

Finally, we want to point to another property of the product, which is the ensemble of AOD solutions for 36 different aerosol types (mixtures of four basic components which are meant to span the realistic range of optical aerosol properties). A user can also propagate all 36 solutions through an application and afterwards calculate the spread of solutions for this ensemble (either evenly distributed which means no knowledge on the aerosol type is assumed) or by using the climatology mix/AOD for the most likely aerosol mix which are also provided in the product files. As performed for calculating the uncertainty due to the uncertain aerosol type, a user can calculate a weighted mean (weighted with the squared distance to the most likely climatology mix in the domain of the three mixing fractions), or a user can select/prescribe one aerosol mix based on available measurements for a case study.

Furthermore, the demonstration dataset also includes a quantitative estimate of uncertainties induced by errors in the cloud masking after the first spatial aggregation level. For this, a probabilistic cloud mask is used to calculate AOD differences with two different cloud probability thresholds for a weak and a strict cloud masking—these thresholds were experimentally optimized to allow a reasonable trade-off between sufficient coverage and (in the final output product) sufficiently strict cloud-masking.

We also include a simple estimate of uncertainties due to sampling (the product of half of the difference between minimum and maximum of input AOD values and the fraction of missing pixels in a grid cell).

More work is needed to assess many details of the uncertainty propagation. Overall, uncertainties seem to be slightly over-estimated which may be due to slightly over-estimated input uncertainties or to missed correlations between two effects (reflectance inversion, albedo estimation), which are based on inverting the same look-up tables. In particular, two of all the inputs make the largest contributions to the final uncertainties on all levels: the uncertainty of the significantly uncertain albedo estimation in the retrieval (and in particular, its common part, which is not reduced by averaging); and the common contribution of the L1B input dataset, which is by far the largest part of the total L1B uncertainty and, again, is not reduced by averaging; it may be possible that the relative part of this common contribution is too large.

## 5. Conclusions

We show here a demonstration dataset of the aerosol optical depth (AOD) over land over Europe and North Africa inferred from the AVHRR instrument onboard two subsequent platforms (NOAA-16 and NOAA-18). The main purpose of this demonstration dataset is to illustrate the propagation of uncertainties benefitting from the methodology and the easyFCDR L1B dataset, which were both developed in the Horizon2020 project FIDUCEO. Despite the very large scatter of the pixel results, we can show good skill of a 10-year climate data record for Europe which agrees well in its seasonality and patterns with a community-merged data record within the ranges of the propagated AVHHR AOD uncertainties.

The significant strength of this demonstration dataset is that its uncertainties could be propagated taking into account detailed correlation structures. The FIDUCEO systematic methodology to analyse the processing chain and the propagation of uncertainties through it proved very useful to structure this analysis. We validate the distributions of propagated uncertainties at different product levels against best estimates of the true error to prove good agreement and a reduction in uncertainties through the processing-chain levels (L2A, L2B, L3_daily) by averaging.

We want to point to specific properties of the product, which are the ensemble of AOD solutions for 36 different aerosol types, a simple estimate of uncertainties due to sampling, and a quantitative estimate of uncertainties induced by errors in the cloud masking.

In conclusion, the new FIDUCEO easyFCDR level 1B input product with its reflectance uncertainties separated into three components with different correlations structures (independent random, common globally correlated, and structured regionally correlated) enabled a mathematically stringent propagation of uncertainties to gridded climate data records of AOD.

**Author Contributions:** T.P. conducted the L2 uncertainty analysis and propagation work while J.M. provided the easyFCDR level 1b input product with its unique uncertainty characterizations and guided on its appropriate use. All authors have read and agreed to the published version of the manuscript.

**Funding:** This research was funded by the European Union's Horizon 2020 research and innovation project FIDUCEO (Fidelity and uncertainty in climate data records from Earth Observations, grant agreement No. 638822). This study was based on the FIDUCEO recipes for analyzing satellite record processing chains for uncertainty contributions and the provision of easyFCDR l1b data input with uncertainties split into components with different correlation structures. In addition, the European Space Agency, as part of the Aerosol_cci+ project (ESA Contract No. 4000126239/19/I-NB), supported the further development of methodology for the validation of pixel-wise uncertainty values by comparing error histograms with histograms from super-imposed Gaussian error distributions per uncertainty value, with this extending state-of-the-art aerosol uncertainty validation.

**Data Availability Statement:** The 10-year AVHRR AOD data record can be downloaded at Dataset Record: FIDUCEO: Advanced Very-High-Resolution Radiometer (AVHRR) Climate Data Record for Aerosol Optical Depth, V1.0, 2003–2012 (ceda.ac.uk, last accessed on 30 December 2021). The underlying input FIDUCEO L1B easyFCDR data record can be downloaded at Dataset Record: FIDUCEO: Fundamental Climate Data Record of recalibrated brightness temperatures for the Advanced Very-High-Resolution Radiometer (AVHRR) with metrologically-traceable uncertainty estimates, 1998–2016, v1.0 (ceda.ac.uk, last accessed on 30 December 2021).

**Acknowledgments:** We are sincerely thankful to Emma Wolliams/NPL for intensive discussions and very helpful guidance in applying the metrological principles to our data. We also are highly grateful to Chris Merchant/Reading university, who led the FIDUCEO project and gave important recommendations on the overall concept and terminology.

**Conflicts of Interest:** The authors declare no conflict of interest. The funders had no role in the design of the study; in the collection, analyses, or interpretation of data; in the writing of the manuscript, or in the decision to publish the results.

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
