# Peer review of "Systematic Propagation of AVHRR AOD Uncertainties—A Case Study to Demonstrate the FIDUCEO Approach"

_remotesensing, doi:10.3390/rs14040875_

Round 1

Reviewer 1 Report

This manuscript gives a very detailed analysis of the uncertainties of AVHRR AOD. Though the calculated uncertainties are not perfectly matching the measurements, they lie in reasonable ranges. I wonder is there any way to estimate or quantify the uncertainty of AOD by the data user? If any, please present. I don't have any serious concerns and there is no problem regarding the structure of this manuscript.

Minor:

  1. LN 90: In "the scattering angle of psi (underline)", remove the underline.
  2. LN 297, Figure 2: in left-bottom, the equations for the 3 uncertainties (common, independent, structured) should take the square root of the right sides.
  3. LN 364: "from eq. 1". The notation of the equation is different from previous equations (e.g. "equation (2)" in LN 241).

Author Response

Way to estimate or quantify the uncertainty of AOD by the data user: We state in lines 646ff that “On all levels of the AVHRR AOD product the user receives total uncertainties (which can be used directly for example in a data assimilation scheme or to ascertain a confident range around values).” And furtheron in this paragraph we give more guidance to a user how he can apply the uncertainty information which is contained in the data files.

Minor:

  1. LN 90: In "the scattering angle of psi (underline)", remove the underline. Done
  2. LN 297, Figure 2: in left-bottom, the equations for the 3 uncertainties (common, independent, structured) should take the square root of the right sides. This is true – We have corrected it in Fig. 2
  3. LN 364: "from eq. 1". The notation of the equation is different from previous equations (e.g. "equation (2)" in LN 241). Thanks for notifying – We have this line accordingly.

Reviewer 2 Report

This study demonstrates the application of the FIDUCEO approach, designed to estimate uncertainty propagations during aerosol AOD retrieving procedures, to estimate the uncertainties at various levels of AVHRR aerosol products. Though not designed for aerosol AOD retrieval, the AVHRR products have the advantage of long continuity since 1980s, which makes it highly valuable to have a thorough investigation of the quality of the AVHRR AOD products. This manuscript does provide such an investigation. The results show a reasonable uncertainty range that is consistent among higher and lower levels of the produces, as well as its implication to absolute errors, hence verify the applicability of the AVHRR AOD products for climatological usage. The method it demonstrated is also a good reference for future followers who would like to apply the AVHRR AOD products and customize the uncertainty estimates according to their particular problems.

The manuscript is very well written, with a clear and detailed description of its algorithms, as well as a robust evaluation of uncertainties over the targeted region. Nevertheless, some minor improvement could be conducted.

  1. Some abbreviations were not given their long terms, such as ‘NDVI’ and ‘CDR’.
  2. The texts and equations in Figure 2 are too small to read easily. It would be better to enlarge them to increase readability. The same is true for Figures 3 and 4.
  3. It’s not mentioned what the colored boxes in Tables 1-5 represent. It seems they are just a visualization strategy without true meaning attached. If that is true, it only adds unnecessary confusion.
  4. Line 467: ‘estimated from physical understanding aerosol plumes …’ should be ‘estimated from physical understanding of aerosol plumes …’
  5. Lines 489-490: It would be helpful to show a map of the targeted region.
  6. Line 580: Why a different rectangle of region is used in this subsection, versus the one depicted at the beginning of section 3 (lines 489-490)?

Author Response

  1. Some abbreviations were not given their long terms, such as ‘NDVI’ and ‘CDR’. We have added those spelled out in lines 144 and 195, resp.
  2. The texts and equations in Figure 2 are too small to read easily. It would be better to enlarge them to increase readability. The same is true for Figures 3 and 4. This is a matter of the final layout of the manuscript. All figures have been provided in high resolution and can be set as full width into the paper. We suggest to address this during final editing.
  3. It’s not mentioned what the colored boxes in Tables 1-5 represent. It seems they are just a visualization strategy without true meaning attached. If that is true, it only adds unnecessary confusion. The coloring adopts the colors used in equations (2) and (3) and Figures 1 and 2 as announced in line 209. We have added a repetition of this coloring explanation in line 329.
  4. Line 467: ‘estimated from physical understanding aerosol plumes …’ should be ‘estimated from physical understanding of aerosol plumes …’ We have added this correction.
  5. Lines 489-490: It would be helpful to show a map of the targeted region. We have added a new Fig. 5 to show the area covered by this demonstration case study (section 3.1) and the sub area covered by the comparison to Sogacheva, et al., 2020 section 3.2).
  6. Line 580: Why a different rectangle of region is used in this subsection, versus the one depicted at the beginning of section 3 (lines 489-490)? The comparison dataset (Europe) is a subset from the larger area for the whole FIDUCEO demonstration dataset (Europe and Northern Africa). The comparison is made for exactly the same area as the one in the merged dataset (Sogacheva et al., 2020) – we have added this explanation in line 586.

Reviewer 3 Report

Dear Authors,

thank you for presenting this interesting manuscript. I have found only some minor issues that I feel should be addressed before the publication:

line 76: "which limit the information content for aerosols to just the total loading" - Even data from two channels should in principle give some information on aerosol mean sizes (through Angstrom exponent). Is such an analysis hindered by relatively long wavelengths available for AVHRRs?

line 88: "(which gives the scattering part...)" - ω0  is missing in the equation and the second σe should be σa instead.

line 96: "AOD550" - why 550? Is this number representative of a wavelength in nm? It is not clear why this number is chosen here.

line 98 "Trace gases contribute no scattering" - I do not believe this is true. Some trace gases do absorb at certain wavelengths but it doesn't mean they have a single scattering albedo equal to zero. What You can say is that their contribution to Rayleigh scattering is negligible because of relatively low concentrations.

line 109: "aerosol type" - this is imprecise. The most important microphysical aerosol parameter for the conversion is the size distribution, not the type (as in chemical composition)

line 146: "BRDF "-  Do these parametrizations take into account the changes in the vegetation during the course of a year?

Table 6: Description of altitude effect - The description seems to be cut/incomplete

Author Response

line 76: "which limit the information content for aerosols to just the total loading" - Even data from two channels should in principle give some information on aerosol mean sizes (through Angstrom exponent). Is such an analysis hindered by relatively long wavelengths available for AVHRRs? In fact, over land the second visible channel (at ~860 mn) lies in the spectral range where vegetation and bare soil have large and significantly varying surface reflectances, so that this channel does not qualify for a dark field inversion method. We have added “over land” in line 77 to make this clear.

line 88: "(which gives the scattering part...)" - ω0  is missing in the equation and the second σe should be σa instead. True. We have added ω0 and corrected the second to σa.

line 96: "AOD550" - why 550? Is this number representative of a wavelength in nm? It is not clear why this number is chosen here. We describe in lines 107 to 110 that the inversion is made at 630 nm and then converted to the commonly used wavelength of 550 nm. We agree that in line 96 it is more consistent to keep it wavelength-independent (as for RTOA) – we have therefore deleted “550” in line 96.

line 98 "Trace gases contribute no scattering" - I do not believe this is true. Some trace gases do absorb at certain wavelengths but it doesn't mean they have a single scattering albedo equal to zero. What You can say is that their contribution to Rayleigh scattering is negligible because of relatively low concentrations. Ok, we have reformulated this sentence following your advice.

line 109: "aerosol type" - this is imprecise. The most important microphysical aerosol parameter for the conversion is the size distribution, not the type (as in chemical composition) This makes sense – we have adapted this statement accordingly – see also line 115, where we define what we call aerosol type in this paper. Note that there are different (and partly inconsistent) practices in different communities for the use of the term aerosol type (in chemical composition or as combination of a set or aerosol properties).

line 146: "BRDF "-  Do these parametrizations take into account the changes in the vegetation during the course of a year? No, this would need further refinements of the parametrizations which hare beyond this case study. To make this clear, we have added “for the entire year” in line 148.

Table 6: Description of altitude effect - The description seems to be cut/incomplete. We have completed the sentence starting with “The“.